Distribution, evolution and expression of GATA-TFs provide new insights into their functions in light response and fruiting body development of Tolypocladium guangdongense

Zhang Chenghua
Wang Gangzheng
Deng Wangqiu dengwq@gdim.cn
Li Taihui mycolab@263.net
State Key Laboratory of Applied Microbiology Southern China, Guangdong Provincial Key Laboratory of Microbial Culture Collection and Application, Guangdong Institute of Microbiology, Guangdong Academy of Sciences , Guangzhou , China
Góes-Neto Aristóteles
Electronic publication date: 2020 Aug 28
Publication date: 2020
Volume: 8
Electronic Location ID: e9784
Received 2020 Feb 27; Accepted 2020 Jul 30
Copyright: ©2020 Zhang et al.
Copyright year: 2020
Copyright holder: Zhang et al.
License: This is an open access article distributed under the terms of the Creative Commons Attribution License, which permits unrestricted use, distribution, reproduction and adaptation in any medium and for any purpose provided that it is properly attributed. For attribution, the original author(s), title, publication source (PeerJ) and either DOI or URL of the article must be cited.
License URL: https://creativecommons.org/licenses/by/4.0/

Keywords: Edible-medicinal fungi, Transcription factor, Photoreceptors, Phylogenetic analysis, Primordial formation

Funding: National Natural Science Foundation of China 31800012 Science and Technology Planning Project of Guangzhou 201804020018 Science and Technology Planning Project of Guangdong Province, China 2018A0303130164 GDAS’ Special Project of Science and Technology Development 2020GDASYL-20200104011 This work was funded by grants from the National Natural Science Foundation of China (Project No. 31800012) and the Science and Technology Planning Project of Guangzhou (No. 201804020018), the Science and Technology Planning Project of Guangdong Province, China (No. 2018A0303130164), and GDAS’ Special Project of Science and Technology Development (No. 2020GDASYL-20200104011). The funders had no role in study design, data collection and analysis, decision to publish, or preparation of the manuscript.

==============================
Background

Fungal GATA-type transcription factors (GATA-TFs) are a class of transcriptional regulators involved in various biological processes. However, their functions are rarely analyzed systematically, especially in edible or medicinal fungi, such as Tolypocladium guangdongense, which has various medicinal and food safety properties with a broad range of potential applications in healthcare products and the pharmaceutical industry.

Methods

GATA-TFs in T. guangdongense (TgGATAs) were identified using InterProScan. The type, distribution, and gene structure of TgGATAs were analyzed by genome-wide analyses. A phylogenetic tree was constructed to analyze their evolutionary relationships using the neighbor-joining (NJ) method. To explore the functions of GATA-TFs, conserved domains were analyzed using MEME, and cis-elements were predicted using the PlantCARE database. In addition, the expression patterns of TgGATAs under different light conditions and developmental stages were studied using qPCR.

Results

Seven TgGATAs were identified. They were randomly distributed on four chromosomes and contained one to four exons. Phylogenetic analysis indicated that GATA-TFs in each subgroup are highly conserved, especially for GATA1 to GATA5. Intron distribution analyses suggested that GATA1 and GATA3 possessed the most conserved gene structures. Light treatments induced the expression levels of TgGATA1 and TgGATA5-7, but the expression levels varied depending on the duration of illumination. The predicted protein structures indicate that TgGATA1 and TgGATA2 possess typical light-responsive domains and may function as photoreceptors to regulate downstream biological processes. TgGATA3 and TgGATA5 may be involved in nitrogen metabolism and siderophore biosynthesis, respectively. TgGATA6 and TgGATA7 possess unique Zn finger loop sequences, suggesting that they may have special functions. Furthermore, gene expression analysis indicated that TgGATA1 (WC1) was notably involved in mycelial color transformation, while other genes were involved in fruiting body development to some extent. These results provide valuable information to further explore the mechanisms through which TgGATAs are regulated during fruiting body development.

Introduction

GATA transcription factors (GATA-TFs) are a class of transcriptional regulators present in fungi, animals, and plants (Patient & McGhee, 2002). Fungal GATA-TFs encode a type IV zinc-finger protein and contain one or two zinc finger domains (CX2CX17-20CX2C), which bind to the DNA sequence (A/T)GATA(A/G) (Lowry & Atchley, 2000; Park et al., 2006; Chi et al., 2013). According to the sequence of CX2CX17-20CX2C domain, fungal GATA-TFs can be grouped into two classes. The first class is the “animal-like” GATA-TFs, which has a leucine in the seventh position of the Zn finger loop, allowing for hydrophobic contact with the first base of (A/T)GATA(A/G) (Scazzocchio, 2000). In some fungi, the “animal-like” GATA-TFs contain two GATA-type DNA-binding domains, also called ‘vertebrate-like’ GATA-TFs, wherein the carboxy-terminal domain normally acts as the DNA-binding finger (An et al., 1997; Patient & McGhee, 2002; Lowry & Atchley, 2000). The second category is the ‘plant-like’ GATA-TFs, which has a glutamic acid residue at position seven of the Zn finger loop (Ballario et al., 1996; Linden & Macino, 1997). This category of GATA-TF generally has a typical PAS domain correlating with light response (Ballario et al., 1996). In addition, in some fungi, there are several other types of GATA-TFs with discrepant sequence features. For instance, in the genome of Saccharomyces cerevisiae, ASH1 is an aberrant GATA-TF with a cysteine residue at the seventh position of the Zn finger loop (Sil & Herskowitz, 1996).

Although the functions performed by GATA-TFs in fungi are very diverse, they generally act as activators or inhibitors to regulate the transcription of a variety of downstream genes, including genes that regulate metabolic processes, cell differentiation and development. These processes are mainly involved in light induction, siderophore biosynthesis, nitrogen metabolism, and mating-type switching (Scazzocchio, 2000; Wong, Hynes & Davis, 2008; Hunter et al., 2014; Niehaus et al., 2017). The ‘plant-like’ GATA-TFs, such as the photoreceptor proteins white collar-1 (WC-1) and white collar-2 (WC-2), are induced by light regulating asexual and sexual development, conidiation, fruiting body development, phototropism, resetting of the circadian rhythm, mycelial carotenoid and sterigmatocystin biosynthesis (Rodriguez-Romero et al., 2010). Ltf1, another ‘plant-like’ member of GATA-TF in Botrytis cinerea (BcLtf1), is also a photosensitive protein, and regulates the light-dependent differentiation, oxidative stress, and the secondary metabolism (Schumacher et al., 2014). NsdD in Aspergillus nidulans, the homologous protein of BcLtf1, is an activator of sexual development and a key repressor of conidiation (Han et al., 2001; Lee et al., 2014). The ‘animal-like’ GATA-TFs AreA and AreB in Fusarium fujikuroi participate in the regulation of nitrogen metabolism (Mihlan et al., 2003; Michielse et al., 2014; Pfannmüller et al., 2017). ASD4, another ‘animal-like’ GATA-TF, is a major transcription regulator in the specification of the lineage of asci and ascospores during sexual development in Neurospora crassa (Feng, Haas & Marzluf, 2000). Urbs-1 in the basidiomycete Ustilago maydis and SREA in the Ascomycete A. nidulans, are both ‘vertebrate-like’ GATA-TFs that act as inhibitors, repressing siderophore biosynthesis (An et al., 1997; Lee et al., 2013). In addition, a newly found GATA-TF Ssams2 in Sclerotinia sclerotiorum, which has a threonine residue in the seventh position of the Zn finger loop, is involved in appressoria formation and chromosome segregation (Liu et al., 2018a; Liu et al., 2018b). However, among the GATA-TFs mentioned above, the light-responsive WC-1 and WC-2, and the nitrogen regulators AreA and AreB, play global roles in fungal growth and development across different species (Niehaus et al., 2017; Pfannmüller et al., 2017). Other GATA-TFs show variable functions in different species, as well as in different stages of development.

So far, there have been many reports on the number and functions of GATA-TFs in plants and animals, but only a few studies have systematically analyzed the functions of GATA-TFs in fungi. In lower eukaryotes such as S. cerevisiae, the family of GATA-TFs contains over 10 members, and the functions of some members are well known (Lowry & Atchley, 2000; Ronsmans et al., 2019). In some plant pathogenic fungi, such as B. cinerea (Schumacher et al., 2014), F. fujikuroi (Niehaus et al., 2017), S. sclerotiorum (Liu et al., 2018a; Liu et al., 2018b; Li et al., 2018), and Magnaporthe oryzae (Quispe, 2011), several GATA-TFs have been successively identified. Several orthologous proteins have also been identified in A. nidulans (Han et al., 2001; Lee et al., 2013; Lee et al., 2014). In macrofungi, research on GATA-TFs in N. crassa is relatively extensive, with six GATA-TFs being identified and characterized (Feng, Haas & Marzluf, 2000; Chen et al., 2009); whereas in some edible or medicinal fungi, the functions of ‘plant-like’ GATA-TFs have been scarcely reported, including the WC-1 and WC-2 in Cordyceps militaris (Yang & Dong, 2014), Ophiocordyceps sinensis (Yang, Xiong & Dong, 2013), Tuber borchii (Ambra et al., 2004), and Lentinula edodes (Sano et al., 2009). In general, the identification and functional analyses of GATA-TFs in edible or medicinal fungi are scattered or insufficiently studied.

Light is an important environmental signal for sexual and asexual growth, circadian rhythm and metabolism in fungi (Kamada et al., 2010; Schumacher, 2017). It is known that two GATA-TFs, WC-1 and WC-2, are directly regulated by light, mediating the induction and repression of light-induced genes (Schumacher, 2012; Canessa et al., 2013). WC-1 interacts with WC-2 to form the White Collar Complex (WCC) so as to activate the transcription of downstream light-regulated genes (Corrochano, 2007; Sano et al., 2009; Sanz et al., 2009); however, their functions in different fungi vary. In most fungi, the WCC is involved in asexual conidiospores production and several metabolic pathways (Rodriguez-Romero et al., 2010), whereas in the human pathogenic fungus Cryptococcus neoformans, the WCC is associated with pathogenicity (Idnurm & Heitman, 2005). In several macrofungi, the WCC is involved in fruiting body development and pigment production (Rodriguez-Romero et al., 2010; Yang et al., 2016). Besides, in the edible and medicinal fungus, C. militaris, the production of pharmacologically active ingredients, such as cordycepin, is also regulated by WC-1 (Yang et al., 2016). Therefore, although WC-1 and WC-2 are identified as photoreceptors in various fungi, it remains unclear how these proteins regulate the developmental and metabolic processes.

Tolypocladium guangdongense, previously known as Cordyceps guangdongsensis, is a type species of the genus Tolypocladium in the Ophiocordycipitaceae family described in the Index Fungorum (http://www.indexfungorum.org/). Similar to Ophiocordyceps sinensis, this fungus has various medicinal properties, and its fruiting bodies are safe and non-toxic (Zhang et al., 2018a; Zhang et al., 2018b; Zhang et al., 2019a; Zhang et al., 2019b). Now it has been considered to be an edible-medicinal fungus, with a broad range of potential applications in health-care products and the pharmaceutical industry. Hence, the mechanism of fruiting body development and biological activities of this species have attracted widespread attention. It is known that GATA transcription factors play vital roles in fungal growth and metabolite synthesis. However, there are very few reports on their characteristics and functions in T. guangdongense. In this study, the essential features of GATA-TFs in T. guangdongense (TgGATAs) were identified and characterized, including the gene structures, amino acid sequences, phylogenetic relationships, and intron distribution. Furthermore, based on the analyses of conserved motifs, promoters, and gene expression under different light conditions and fruiting body developmental stages, we investigated the functions of TgGATAs.

Materials & Methods

Identification of TgGATAs

To identify the TgGATAs, the online analysis platform, InterProScan (Zdobnov & Apweiler, 2001) was used to screen the T. guangdongense protein database for proteins with conserved GATA-domains (Zhang et al., 2018a; Zhang et al., 2018b). Each GATA-TF candidate sequence was further confirmed by domain analyses using the Pfam protein family database (http://pfam.xfam.org) (Finn et al., 2014) and SMART databases (http://smart.emblheidelberg.de/) (Letunic, Doerks & Bork, 2012).

Sequence analysis of TgGATAs

The characteristics of the GATA-domains in TgGATAs were analyzed using the ClustalW and BioEdit program (Thompson et al., 1997; Hall, 1999). The DNA and CDS sequences of the predicted GATA-TFs were obtained from the T. guangdongense genome. Exon/intron structures were obtained by comparing the cDNA sequences obtained in the previous study (Zhang et al., 2019a; Zhang et al., 2019b) and the corresponding genomic DNA sequences (Zhang et al., 2018a; Zhang et al., 2018b). The isoelectric point (pI) and molecular weight (Mw) of the TgGATAs were calculated using ExPASy tool (http://web.expasy.org/compute pi/). The subcellular localization of each protein was analyzed using BaCelLo Prediction (http://gpcr.biocomp.unibo.it/bacello/pred.htm). The location of the TgGATAs on the chromosomes was determined using the T. guangdongense genome database.

Homologous protein identification and phylogenetic analysis of the GATA family

The homologous proteins of GATAs were identified by genome-wide BLAST analyses. Multiple alignments of GATA-TF protein sequences were performed using the ClustalW program (Thompson et al., 1997). Phylogenetic trees were constructed to analyze the sequence similarity of the GATA factors using the neighbor-joining (NJ) method with a Kimura 2-parameter model in MEGA 5.0 (Tamura et al., 2011). The stability of the internal nodes was assessed by bootstrap analysis of 1,000 replicates. The phylogenetic tree was visualized using iTOL (http://itol.embl.de/help.cgi). Homologous proteins were identified by conducting a BLASTP search (McGinnis & Madden, 2004) against forty fungal genomes, which were obtained from the GenBank database or Ensembl Fungi database.

Gene structures and intron analysis of TgGATAs

The online gene structure display server (GSDS) (http://gsds.cbi.pku.edu.cn/) was used to generate the exon-intron structures of the GATA-TFs, including the exon positions and gene lengths (Hu et al., 2015). The position of the introns was determined based on the genome sequences of the selected species.

Conserved motif analysis of TgGATAs

The conserved motifs of the GATA-TFs were studied using the online MEME program (http://meme-suite.org/). Analyses were performed using the following set of parameters: the maximun motif width was set to 50 and the maximum number of motifs was set to 10. Only the motifs with P values <10−6 and not overlapping with each other were reported. The secondary structures of the GATA-TFs were identified using the SMART databases.

Cis-elements analysis of TgGATAs

Promoter sequences comprising the 1,500 bp upstream sequences of the GATA-TF genes were derived from the transcription start site based on the T. guangdongense genome (Zhang et al., 2018a; Zhang et al., 2018b). The conserved cis-acting regulatory elements present in the promoter regions of the identified GATA-TF gene sequences were computationally predicted using the PlantCARE database (http://bioinformatics.psb.ugent.be/webtools/plantcare/html/) (Lescot et al., 2002). Cis-acting regulatory elements responsive to light were manually searched. Any gene whose promoter region contained a certain light-responsive element was noted.

Expression analysis of TgGATAs under different conditions

For the studies on relative gene expression in response to light, the Tolypocladium guangdongense strain GDGM30035 was cultured on PDA medium at 22 ± 1 °C under dark conditions. After 20 days (the colony diameter was approximately half of the diameter of the 90 cm culture dish), samples were exposed to light (1,600 lux) for 15 min, 30 min, 2 h, and 4 h, respectively. Strains cultured under dark conditions were used as controls. Samples were immediately collected at the indicated time points. For the studies on relative gene expression during fruiting body development, the strain was inoculated on liquid YMPD medium (per liter: 3 g of yeast extract, 10 g of glucose, 2 g of malt extract, and 5 g of peptone) and cultured at 22 ± 1 °C with shaking under dark conditions. After 10 days, mycelia were inoculated on rice media (per tissue culture vessels: 25 g rice, and 20 ml nutrient solution). The composition of the nutrient solution was as follows: 20 g of sucrose, 5 g of beef extract, 10 g of KNO3, 4 g of soybean, and pH 6.0–6.5 per liter. The samples were cultured under dark conditions for two weeks and then shifted to alternating light and dark conditions (10L/14D) based on a previous study (Lin et al., 2009). Samples for RNA extraction were collected at different developmental stages, including vegetative stage (M, the mycelia color is white), color transition period (TC, the mycelia color changes from white to yellow), primordium (P, the surface of the hyphae forms primordium), early stage of fruiting body development (FB1, the length of fruiting body is approximately 1–2 cm), middle stage of fruiting body development (FB2, the length of fruiting body is approximately 3–4 cm), late stage of fruiting body development (FB3, the length of fruiting body is longer than 5 cm in size), and mature fruiting body (FM, the length of fruiting body is longer than 5 cm in size and atrovirens). Total RNA was extracted using Trizol, and 1 µg of each RNA sample was used for reverse transcription with the HiScript II Q RT SuperMix (+gDNA wiper) (Vazyme). Real-time PCR was conducted in a CFX384 real-time system (Applied Biosystems) using TakaRa SYBR Premix ExTaq (TakaRa Biotechnology Co.) with specific primers (Table S1), and the following parameters: initial preheating at 95 °C for 30 s, followed by 39 cycles at 95 °C for 5 s, and 60 °C for 30 s. The vacuolar protein sorting (VPS) and histone H4 (H4) genes were selected as reference genes for analysis the relative expression levels of TgGATAs at different developmental stages. Based on the previous transcriptome analyses under different illumination times, eight genes (actin, α-tub1, α-tub2, β-tub1, rpb, EF1- β, VPS, and H4) described by Wang et al. (2020) were selected as candidate genes to analyze the expression stability under different light conditions by two statistical algorithms (geNorm and NormFinder). Gene expression levels were calculated using the 2−ΔΔCt method (Livak & Schmittgen, 2001).

Results

Identification and characterization of GATA-TFs

Based on the genome sequence of T. guangdongense, seven predicted genes were identified as candidates, accounting for 1.61% of the total predicted TFs in the T. guangdongense genome. After further confirmation by domain analysis using the Pfam protein family database and SMART database, the seven identified GATA-TFs were denoted as TgGATA1 to TgGATA7 (Table S2). All the proteins comprised one Cys4 (C4) Zn finger domain, except for TgGATA5, which had two C4 Zn finger domains (Fig. 1). In the comparison of the C4 Zn finger domain sequences in different GATA-TFs, four proteins (TgGATA3, TgGATA4, TgGATA5, and TgGATA7) possessed a well-conserved GATA motif, type IVa (Cys-X2-Cys-X17-Cys-X2-Cys), while the remaining proteins possessed another well-conserved GATA motif, type IVb (Cys-X2-Cys-X18-Cys-X2-Cys). Based on the amino acid residues in the seventh position of the Zn finger loops, TgGATA1, TgGATA2, and TgGATA6 belonged to the ‘plant-like’ GATA-TFs, while TgGATA3, TgGATA4, TgGATA5 and TgGATA7 were classified as ‘animal-like’ GATA-TFs. Furthermore, TgGATA5 could also be categorized as a ‘vertebrate-like’ GATA-TF due to the presence of two GATA-type DNA-binding domains.

Figure 1 GATA-domain analysis of the identified GATA-TFs in Tolypocladium guangdongense.

The threshold for consensus highlighting was 30%. The black asterisk denoted consensus sequences, and red star represents the seventh position of the Zn finger loop.

Detailed information on GATA-TFs is listed in Table 1, including the gene ID, chromosome location, protein length, molecular weight (Mw), the oretical isoelectric point (pI), and prediction of subcellular location. The seven proteins were mapped to four chromosomes in the T. guangdongense genome. TgGATA1, TgGATA4 and TgGATA7 were present on chromosome 1, and the first two had the same orientation, while the third one had opposite orientations. TgGATA2 and TgGATA6 were present on chromosomes 2 and 3, respectively. TgGATA3 and TgGATA5 were present on chromosome 4 with the same orientation.Proteins encoded by the predicted GATA-TFs ranged from 195 to 1,034 amino acids in length, with an average size of 588 amino acids. The predicted molecular weights of these proteins ranged from 40.49 to 112.10 kDa (average 63.20 kDa). All the predicted proteins had an isoelectric point (pI) below 10, except for TgGATA6 that had a pI of 11.08. Based on the analysis of the predicted subcellular localization, only one protein (TgGATA7) was predicted to be localized to the mitochondria, and the rest were predicted to be localized to the nucleus. Nuclear localization signal sequence analyses showed that three proteins had the bipartite-type nuclear localization sequences, including TgGATA1 (RKESRPEFGRAIEKARR), TgGATA3 (RRHRKTSIDERRNRKRP) and TgGATA6 (RRDSPSADADASGRSRR).

Table 1 Characterization of GATA-TFs in Tolypocladium guangdongense.

Gene name	Gene ID	Location	aa	Exon	PI	Mw(Da)	Subcellular location	
TgGATA1	CCG_01872	Chromosome 1	1034	2	7.3	112095.44	Nucleusa	
TgGATA2	CCG_04254	Chromosome 2	541	3	5.51	59186.27	Nucleus	
TgGATA3	CCG_07987	Chromosome 4	979	3	9.24	102733.69	Nucleusa	
TgGATA4	CCG_02574	Chromosome 1	376	4	5.54	40488.91	Nucleus	
TgGATA5	CCG_07433	Chromosome 4	535	3	9.07	56338.17	Nucleus	
TgGATA6	CCG_06499	Chromosome 3	195	3	11.08	20884.24	Nucleusa	
TgGATA7	CCG_00651	Chromosome 1	462	1	7.22	50682.75	Mitochondrion	
Notes.

a Nuclear localization signal sequences found in amino acid sequences. Detailed information of the TgGATAs are provided in Table S2.

Table 2 Comparison of GATA-TFs between Tolypocladium guangdongense and other fungi.

	Family	Species	GATAsa	Other GATAsb	
Ascomycota			1	2	3	4	5	6	7	Number	Referencec	
Dothideomycetes												
Pleosporales	Leptosphaeriaceae	Leptosphaeria maculans	✓	✓	✓	/	✓	/	/	/	–	
Capnodiales	Mycosphaerellaceae	Zymoseptoria tritici	✓	✓	✓	✓	✓	/	/	/	–	
Eurotiomycetes												
Eurotiales	Aspergillaceae	Aspergillus fumigatus	✓	✓	✓	✓	✓	✓	✓	/	Nierman et al. (2005)	
Eurotiales	Aspergillaceae	Aspergillus flavus	✓	✓	✓	✓	✓	✓	✓	/	–	
Eurotiales	Aspergillaceae	Aspergillus nidulans	✓	✓	✓	✓	✓	✓	/	1	Lowry & Atchley (2000)	
Eurotiales	Aspergillaceae	Aspergillus niger	✓	✓	✓	✓	✓	✓	/	/	Park et al. (2006)	
Eurotiales	Aspergillaceae	Aspergillus terreus	✓	✓	✓	✓	✓	/	/	/	FTFD	
Eurotiales	Aspergillaceae	Penicillium chrysogenum	✓	✓	✓	✓	✓	/	/	5	FTFD	
Eurotiales	Aspergillaceae	Penicillium marneffei	✓	✓	✓	✓	✓	✓	/	5	FTFD	
Eurotiales	Aspergillaceae	Talaromyces stipitatus	✓	✓	✓	✓	✓	/	/	6	FTFD	
Onygenales	Ajellomycetaceae	Histoplasma capsulatum	✓	/	✓	✓	/	/	/	2	Park et al. (2006)	
Leotiomycetes												
Helotiales	Drepanopezizaceae	Marssonina brunnea	✓	✓	✓	✓	✓	/	/	/	–	
Helotiales	Helotiaceae	Glarea lozoyensis	✓	✓	✓	✓	✓	/	/	/	–	
Helotiales	Ploettnerulaceae	Rhynchosporium commune	✓	✓	✓	✓	✓	/	/	/	–	
Thelebolales	Thelebolaceae	Pseudogymnoascus destructans	✓	✓	✓	✓	✓	/	/	/	–	
Helotiales	Sclerotiniaceae	Botrytis cinerea	✓	✓	✓	✓	✓	✓		1	Park et al. (2006)	
Helotiales	Sclerotiniaceae	Sclerotinia sclerotiorum	✓	✓	✓	✓	✓	✓	/	2	Park et al. (2006), Li et al. (2018), Liu et al. (2018a) and Liu et al. (2018b)	
Pezizomycetes												
Pezizales	Tuberaceae	Tuber borchii	✓	✓	✓	✓	✓	/	/	/	Ambra et al. (2004)	
Pezizales	Tuberaceae	Tuber melanosporum	✓	✓	✓	✓	✓	/	/	/	–	
Saccharomycetes												
Saccharomycetales	Dipodascaceae	Yarrowia lipolytica	✓	/	✓	✓	✓	/	/	5	Dujon et al. (2004)	
Saccharomycetales	Saccharomycetaceae	Saccharomyces cerevisiae	N	N	/	✓	/	/	/	10	Lowry & Atchley (2000), Park et al. (2006)	
Sordariomycetes												
Glomerellales	Glomerellaceae	Colletotrichum graminicola	✓	✓	✓	✓	✓	/	/	/	–	
Hypocreales	Cordycipitaceae	Cordyceps militaris	✓	✓	✓	✓	✓	/	/	/	–	
Hypocreales	Hypocreaceae	Trichoderma atroviride	✓	✓	✓	✓	✓	/	/	2	FTFD	
Hypocreales	Hypocreaceae	Trichoderma reesei	✓	✓	✓	✓	✓	/	/	2	Park et al. (2006)	
Hypocreales	Incertae sedis	Acremonium chrysogenum	✓	✓	✓	✓	✓	/		/	–	
Hypocreales	Clavicipitaceae	Ustilaginoidea virens	✓	✓	✓	✓	✓	/	/	2	Yu et al. (2019)	
Hypocreales	Nectriaceae	Fusarium fujikuroi	✓	✓	✓	✓	✓	✓	✓	/	Michielse et al. (2014) and Niehaus et al. (2017)	
Hypocreales	Nectriaceae	Fusarium oxysporum	✓	✓	✓	✓	✓	/	/	3	FTFD	
Hypocreales	Ophiocordycipitaceae	Ophiocordyceps sinensis	✓	✓	✓		✓	/	/	/	–	
Hypocreales	Ophiocordycipitaceae	Tolypocladium guangdongense	✓	✓	✓	✓	✓	✓	✓	/	this study	
Hypocreales	Ophiocordycipitaceae	Tolypocladium ophioglossoides	✓	✓		✓	✓	/	/	/	–	
Hypocreales	Ophiocordycipitaceae	Tolypocladium paradoxum	✓	✓	✓	✓	✓	✓	✓	/	–	
Magnaporthales	Magnaporthaceae	Magnaporthe oryzae	✓	✓	✓	✓	✓	/	/	4	Dean et al. (2005); FTFD	
Sordariales	Chaetomiaceae	Chaetomium globosum	✓	✓	✓	✓	/	/	/	1	Park et al. (2006); FTFD	
Sordariales	Sordariaceae	Neurospora crassa	✓	✓	✓	✓	✓	✓	/	1	Borkovich et al. (2004) and Li et al. (2018)	
Basidiomycota												
Agaricomycetes												
Agaricales	Psathyrellaceae	Coprinopsis cinerea	✓	/	/	✓	✓	✓	/	/	–	
Agaricales	Pleurotaceae	Pleurotus ostreatus	✓	/	/	✓	/	/	✓	6	FTFD	
Tremellomycetes												
Tremellales	Tremellaceae	Cryptococcus neoformans	✓	✓	✓	✓	✓	/	/	7	Park et al. (2006)	
Ustilaginomycetes												
Ustilaginales	Ustilaginaceae	Ustilago maydis	✓	✓	✓		✓	/	✓	5	Kamper et al. (2006)	
Notes.

a GATA1-7 represent the corresponding homologues of TgGATA1-7. /, represents that non- homologue or lower homology GATA transcription factor was found.

b Other GATAs represent the non-homologues or lower homology GATAs with TgGATA1-7 in each species.

c The homologues of TgGATA1-7 were identified by genome-wide Blast analyses; FTFD, the total number of GATAs was obtained from the Fungal Transcription Factor Database (FTFD); some of GATAs and the number of GATAs in special species have been reported by previous studies.

Distribution and phylogenetic analyses of the GATA-TFs family in fungi

An overview of the identified GATA-TFs in T. guangdongense and related homologous proteins in different classes of Ascomycota and Basidiomycota is shown in Table 2. The homologous proteins of five GATA-TFs in T. guangdongense (TgGATA1- TgGATA5) were identified by genome-wide analyses. Besides, these proteins were highly conserved in Ascomycota, except for Saccharomyces cerevisiae, suggesting that these homologous proteins in different fungi may be the main functional proteins, and may also have similar functions. Compared to Basidiomycota, GATA1 and GATA4 were relatively conserved, while orthologs of the remaining proteins were randomly distributed across different species. Proteins similar to TgGATA6 and TgGATA7 were found in several species. Other GATA-TFs are also listed in Table 2. Due to the lower homology, whether some of them are paralogs of the seven types of GATA-TFs in some cases needs to be further analyzed.

To analyze the sequence similarity between GATA-TF genes in T. guangdongense and those in other fungi, 143 full-length amino acid sequences in Ascomycota were used to construct an unrooted phylogenetic tree using the neighbor-joining (NJ) method. Among these, 43 sequences were from the Eurotiomycetes class, 29 sequences were from the Leotiomycetes class, and 71 sequences were from the Sordariomycetes class. These GATA-TFs proteins were classified into seven distinct subgroups, with support values over 90%. The higher supporting rates of each subgroup implied a relatively higher level of synteny between the same types of GATA-TF proteins across different species (Fig. 2). Subgroup I was closely related to subgroup II. However, TgGATA1 (CCG_01872) and TgGATA2 (CCG_04254) shared only a 10% amino acid identity. Although TgGATA6 and TgGATA7 showed lower degrees of identity with those sequence similar proteins listed in Table 2, these proteins were clustered into two subgroups (VI and VII) with 100% bootstrap values. All proteins in the GATA-TF subgroups I and II were divided into three branches with bootstrap values within 80%–90%, and belonged to the three classes of Ascomycota, respectively. The proteins in the GATA-TF subgroups III, IV, and VII were clearly clustered into three classes of Ascomycota with bootstrap values within 90%–100%, while the proteins in GATA-TF subgroups V and VI were grouped into three classes with bootstrap values of 100%. These results indicate that all the proteins in the same subgroups are highly conserved among different classes, even in Ascomycota.

Figure 2 Phylogenetic tree of GATA-TFs from Tolypocladium guangdongense and other fungi in three classes of Ascomycota, including Eurotiomycetes, Leotiomycetes and Sordariomycetes.

The phylogenetic tree was constructed using the NJ (neighbor-joining) method with 1,000 bootstrap replications. The different classes are distinguished by different colors. Detailed information on the homologous protein is provided in Table S3.

Gene structures and intron distribution of TgGATAs

In order to gain further insight into the evolutionary relationships of the GATA-TFs in T. guangdongense, the exon-intron structure for each member of this family was analyzed. The number of exons in the GATA-TFs ranged from one to four (Fig. 3). From the comparison of the seven GATA-TFs, the exon-intron structures of GATA2, GATA3, GATA5 and GATA6 were found to be similar, with three exons each. GATA4 had the largest number of exons with four exons, while GATA7 had only one exon. GATA2 had the largest intron sequence among all the GATA-TFs, and GATA1 had the smallest intron sequence.

Figure 3 Intron-exon structure analysis of GATA-TFs in Tolypocladium guangdongense.

Information regarding the intron and exon positions are provided in Table S4.

To determine the relationship between the GATA-TF genes of T. guangdongense and its homologous genes in other fungi, the distribution of GATA-TF introns was investigated (Fig. 4). The intron numbers of TgGATA1 and TgGATA3 were consistent with the homologous genes in Sordariomycetes, with an intron length between 53 bp and 101 bp. The intron positions of GATA1 and GATA3 were also relatively consistent, except for that of GATA1 in Ophiocordyceps sinensis. The introns of GATA1 and GATA3 in the other fungi classes showed random distributions both in terms of the number of introns and their positions. By comparison, significant differences existed in the intron distributions of GATA2 and GATA4 in the same classes, as well as in the same families. However, the intron number of TgGATA4 was consistent with those in Tolypocladium ophioglossoides and Tolypocladium paradoxum, which belong to the same genus. The intron number of GATA5 in the Sordariomycetes class was consistent, except for that in Ophiocordyceps sinensis. These results indicate that GATA1 and GATA3 are more conserved than other members of the GATA-TFs in the Sordariomycetes class.

Figure 4 Intron/exon structures of TgGATA1-5 and their homologous genes in other fungi.

Intron positions in the GATA-TFs of Tolypocladium guangdongense and other fungi are denoted by different colored triangles on the amino acid sequences. Sequences 1 to 13 represent the corresponding orthologs in Tolypocladium guangdongense, Tolypocladium ophioglossoides, Tolypocladium paradoxum, Ophiocordyceps sinensis, Fusarium fujikuroi, Fusarium oxysporum, Cordyceps militaris, Neurospora crassa, Magnaporthe oryzae, Botrytis cinerea, Sclerotinia sclerotiorum, Aspergillus nidulans, Aspergillus niger. Detailed information on the number of amino acids and the intron positions are provided in Table S5.

Motif analysis of TgGATAs

In order to investigate the functions of TgGATAs, their conserved motifs in T. guangdongense and the known functional GATA-TFs in other fungi were identified using the MEME. Ten motifs were identified with lengths varying from 20 to 50 amino acids. Detailed information of the 10 putative motifs is provided in Fig. S1. As shown in Fig. 5, the motif compositions of TgGATA1 and TgGATA2 were similar to those of WC-1 and WC-2 in other fungi, which contained two conserved GATA-type Zn finger motifs, and two conserved PAS motifs. These results indicate that TgGATA1 and TgGATA2 are homologues of WC-1 and WC-2, which function as photoreceptors in response to light. Although the amino acid sequence of TgGATA6 shared a lower degree of identity with the homologues of NsdD, these proteins shared similar conserved motifs. The fact that NsdD and its homologues are regulated by light, and TgGATA6 is also classed as a ‘plant-like’ GATA-TF, suggesting that TgGATA6 is likely to be a downstream target genes activated in response to light. The same dynamics seemed to be found in TgGATA3 and NIT2, as well as in TgGATA5 and SREA, suggesting that TgGATA3 and TgGATA5 may be involved in nitrogen regulation and siderophore biosynthesis, respectively. Both TgGATA4 and TgGATA7 had similar functional motifs as ASD4 in N. crassa (NcASD4), however, the amino acid sequence of TgGATA4 was much closer to that of NcASD4.

Figure 5 Motif analysis of GATA-TFs in Tolypocladium guangdongense and known functional GATA-TFs in other fungi.

The phylogenetic tree was constructed using the NJ (neighbor-joining) method with 1,000 bootstrap replications. Ten conserved motifs were identified in the proteins and are indicated in different colored boxes. ZnF, represented the GATA-type Zinc finger domain. PAS, represented the PAS domain. UN, represented uncharacterized motif. The scale bar indicated the number of amino acids (aa). Each motif sequence and alignment is shown in Table S6 and Fig. S1. An, A. nidulans; Bc, B. cinerea; Cm, C. militaris; Ff, F. fujikuroi; Nc, N. crassa; Os, O. sinensis; Ss, S. sclerotiorum; Tg, T. guangdongense. GenBank numbers of known functional GATA-TFs are listed as follows: AnNsdD, AAB16914; AnSREA, AAD25328; Bcltf1, ANQ80444; BcWC-1, XP_024547291; CmWC-1, EGX96523; Ffcsm1, CCT68588; NcADS4, AAG45180; NcNIT2, P19212; NcSREA, EAA32742; NcSub-1, ESA42507; NcWC-1, Q01371; NcWC-2, EAA34583; OsWC-1, EQK98623; Ssams2, SS1G_03252; Sssfh1, SS1G_01151; SsNsd1, ANQ80447.

In order to confirm the photoreceptor roles of TgGATA1 and TgGATA2, their secondary structures were analyzed using the SMART program (Fig. 6A). Like the other white-collar proteins in C. militaris and O. sinensis, TgGATA1 contained an N-terminal glutamine-rich region, a LOV domain, two PAS domains, and a Zn finger domain. TgGATA2 contains a PAS domain and a Zn finger domain. However, unlike WC-1 in N. crasssa, TgGATA1 lacked a C-terminal glutamine-rich region. Except for a GATA-type Zn finger domain, TgGATA3 also contains an unknown functional domain DUF1752, while the other proteins, TgGATA4-TgGATA7, had no obvious domains.

Figure 6 Structural features and light-responsive cis-element analysis of GATA-TFs in Tolypocladium guangdongense.

A, Schematic representation of GATA-TFs in T. guangdongense. The positions of the LOV domain, the PAS domain and the zinc-finger domain were predicted using the SMART program (http://smart.embl-heidelberg.de/). B, Analyses of cis-elements in promoter regions of TgGATAs. The promoter sequences (−1,500 bp) of TgGATAs were used for analyses. The different types of cis-elements are indicated by various geometric figures at the corresponding positions, and detailed information is listed in Table S7.

Cis-elements and gene expression analyses of TgGATAs under light

To further explore the functions of TgGATAs in response to light, the cis-elements in their promoter regions were predicted (Fig. 6B). In total, ten types of cis-elements were identified, ranging from two to seven in each gene. Among the ten cis-elements, six were responsible for light responsiveness (Box 4, G-box, Sp1, TCT-motif, TCCC-motif, and GATA-motif), three were involved in stress responsiveness (LTR, TC-rich repeats, and MBS), and one was related to circadian control (circadian). In the three ‘plant-like’ GATA-TFs, TgGATA1 possessed seven cis-elements, including four light-responsive cis-elements, two stress-responsive cis-elements, and one circadian control cis-element. It is speculated that the expression of TgGATA1 may be associated with these biological processes. TgGATA2 only possessed two cis-elements, and both were involved in light responsiveness, while TgGATA6 possessed five light-responsive cis-elements, and one stress-responsive cis-element. In the ‘animal-like’ GATA-TFs, all were found to possess different numbers of light-response cis-elements, however, whether their expression was regulated by light remains to be proven. In addition, TgGATA5 also possessed a GATA-motif, suggesting that this gene may interact with other GATA-TFs genes.

To test the above hypotheses, the expression of GATA-TFs in T. guangdongense was further analyzed under different light conditions by quantitative real-time PCR. Based on the results on reference genes, α-tub1 and β-tub1 were selected as reference genes for analysis of relative expression levels of TgGATAs under different light conditions (Table S8). TgGATA1 and TgGATA2 exhibited slight up-regulation when β-tub1 was used as the reference gene (Fig. 7B), however, no difference was observed when α-tub1 was used as the reference gene (Fig. 7A). After light treatment for 30 min, the relative expression level of TgGATA1 decreased, while the relative expression level of TgGATA2 showed a slight up-regulation with no significant difference. The relative expression levels of TgGATA3 and TgGATA4 were down-regulated, but only the expression of TgGATA4 was significantly decreased after light treatment. The relative expression levels of TgGATA5 and TgGATA6 remained up-regulate patterns after light treatment, and their expression levels changed more markedly after light treatment for 30 min. The relative expression level of TgGATA7 also exhibited an up-regulate pattern, however, a significant difference was observed in light treatment for 4 h.

Figure 7 Quantitative real-time PCR analyses of TgGATAs in response to light.

Gene expression was measured after different illumination times. The mean expression value was calculated from three independent replicates. The vertical bars indicate the standard deviation. Expression level was normalized by the selected reference genes α-tub-1 (A) and β-tub-1 (B).

Expression analysis of TgGATAs during fruiting body development

To better understand the functions of GATA-TFs in T. guangdongense, the expression patterns of TgGATAs were studied at the different developmental stages of the fruiting body (Fig. 8). During developmental stages, TgGATA1 was significantly up-regulated at the stage where hyphal color changed from white to yellow (TC). TgGATA2 was obviously up-regulated from the primordium formation stage (P) to the fruiting body developmental stages (FB1 and FB2). The expression level of TgGATA3 significantly increased at the mature fruiting body stage (FM). The relative expression levels of TgGATA4 and TgGATA5 exhibited down-regulation trends mainly from the TC stage to FB2 stages, and changed more obviously at the TC stage. TgGATA6 was significantly up-regulated at the late stage of fruiting body development (FB3) and mature fruiting body stage. TgGATA7 maintained continuous up-regulation from the primordium formation stage to mature fruiting body stage. Although the expression levels of all genes changed in obviously varying degrees at some stages by normalizing with VPS and H4 (Figs. 8A and 8B), the varying trends were consistent.

Figure 8 Quantitative real-time PCR analyses of TgGATAs during fruiting body development.

Gene expression was measured in different developmental stages, including vegetative stage (M), color transition period (TC), primordia (P), early stage of fruiting body development (FB1), middle stage of fruiting body development (FB2), late stage of fruiting body development (FB3), maturing period of fruiting body (FM). The mean expression value was calculated from three independent replicates. The vertical bars indicate the standard deviation. Expression level was normalized by the selected reference genes VPS (A) and H4 (B).

Discussion

GATA-TFs are widely distributed in fungi, animals, and plants (Patient & McGhee, 2002). However, the number of GATA-TFs varies greatly within and between the three kingdoms. In plants, GATA-TFs have been systematically characterized in many species, such as Arabidopsis thaliana (Reyes, Muro-Pastor & Florencio, 2004), Gossypium raimondii, G. arboretum, G. hirsutum (Zhang et al., 2019a; Zhang et al., 2019b), and Vitis vinifera (Zhang et al., 2018a; Zhang et al., 2018b), where the number of GATA-TFs ranges from 19 to 87. In vertebrates, six GATA-TFs have been identified with well-characterized functions in disease control (Lentjes et al., 2016; Tremblay, Sanchez-Ferras & Bouchard, 2018), and 11 GATA-TFs have been identified in Caenorhabditis elegans, (Block & Shapira, 2015). Over ten GATA-TFs have been found in yeast (Ronsmans et al., 2019). While in other fungi, 3 to 16 members of GATA-TFs have been found with the help of genome-wide analyses (Park et al., 2006). In previous studies, very few GATA-TFs have been analyzed in some edible or medicinal fungi, and the total number of GATA-TFs and their functions remain little known. The present study is the first to systematically analyze the GATA-TFs in the edible and medicinal fungus, T. guangdongense. As a result, seven GATA-TFs were identified, indicating that fungi might possess relatively fewer GATA-TFs than that in plants.

According to the domain features, two ‘plant-like’ GATA-TFs, GATA1 (homolog of WC-1) and GATA2 (homolog of WC-2), were classified as photoreceptors, which are widely distributed in Ascomycota (Rodriguez-Romero et al., 2010). However, there is one exception, S. cerevisiae lacks light responses and WC photoreceptors do not exist in this fungus (Park et al., 2006). WC-1 and WC-2 have also been found in Zygomycetes and Basidiomycetes (Corrochano & Garre, 2010; Rodriguez-Romero et al., 2010). However, a homolog of WC-2 was identified in Coprinopsis cinerea and Cryptococcus neoformans with a relatively lower identity compared to those in Ascomycota (Park et al., 2006; Corrochano, 2007; Kuratani et al., 2010), indicating that GATA1 is more conserved than GATA2 in fungi. Protein domain and intron distribution analyses also suggested that GATA-TFs in the GATA1 subgroup shared similar gene structure and protein domain. As WC-1 often interacts with WC-2 to form WCC, we further analyzed the expression correlations of TgGATAs based on previous transcriptome data (Zhang et al., 2019a; Zhang et al., 2019b) by weighted correlation network analysis (WGCNA) according to the method described by Langfelder & Horvath (2008), following the general WGCNA guidelines (Zhang & Horvath, 2005) (Table S9). Co-expression network analysis showed that significant positive regulatory relationships exist between TgGATA1 and TgGATA2, with a pairwise Pearson correlation coefficients of 0.88 (P = 1.13E-08). Additionally, the WGCNA analysis results also suggested that the possible regulatory connections may exist between TgGATA1-TgGATA7 and TgGATA2-TgGATA7, with the pairwise Pearson correlation coefficients of 0.75 (P = 1.82E-05) and 0.82 (P = 9.34E-07), respectively. These results suggested that TgGATA7 may be a downstream target gene of WCC; however, this hypothesis needs to be confirmed in further studies. The ‘animal-like’ GATA-TFs in fungi contained four members, and all of which are likely to be found in both Ascomycota and Basidiomycota. Based on the conserved domains of homologous proteins, TgGATA3 and TgGATA5 may have functions similar to AreA and SreA, which are found to be mainly involved in the regulation of the nitrogen metabolism and siderophore biosynthesis, respectively. AreA acts as a positive regulator of nitrogen metabolite repression (NMR) sensitive genes involved in the utilization of alternative nitrogen sources, while SreA deficiency not only leads to the repression of siderophore biosynthesis but also results in the deregulation of siderophore-bound iron uptake and ornithine esterase expression (Oberegger et al., 2001). TgGATA4 is highly homologous with AreB or ASD4. Apart from being involved in the regulation of the nitrogen metabolism (Michielse et al., 2014), AreB also acts as a strong repressor of bikaverin gene expression (Pfannmüller et al., 2017). In N. crassa, ASD4 is involved in ascus and ascospore development (Feng, Haas & Marzluf, 2000). On the basis of these findings, the association of TgGATA4 with developmental processes or the regulation of metabolic processes needs to be further investigated.

So far, eight characterized GATA-TFs have been identified in fungi, including the nitrogen regulators AreA/NIT2 and AreB (or the sexual development regulator ASD4), the central components of the blue light-sensing system WC-1 and WC-2, the sexual and asexual development regulator NsdD/SUB-1 (Han et al., 2001; Colot et al., 2006; Lee et al., 2014), the iron uptake regulator SreA (Machida & Gomi, 2010), SFH1 (involved in hyphal growth, reactive oxygen species accumulation, and pathogenicity) (Liu et al., 2018a), and AMS2 (associated with appressoria formation and chromosome segregation) (Liu et al., 2018b). In T. guangdongense, homologs of AreA, AreB/ASD4, WC-1, WC-2, and SreA were found, and the alignments of the homologous GATA-domains were very similar with high degrees of identity (Fig. S2A-E). However, no reliable homologs of NsdD/SUB-1, SFH1 and AMS2 were found in T. guangdongense. Although the phylogenetic analysis of TgGATA6 was clustered into NsdD/SUB-1 with an approval rate of 100%, the amino acid sequence of TgGATA6 showed significant differences from those of NsdD/SUB-1 and their homologs. This phenomenon was a little strange, and a probable alternative splicing in this gene under different light conditions may be the cause. Phylogeny and domain analyses also indicated that TgGATA7 was significantly different from the other proteins. Furthermore, alignments of the GATA-domains between the above two proteins and the known functional proteins demonstrated that TgGATA6 possesses similar amino acid sequences of the Zn finger loop with those in AnNsdD, SsNsd1, BcLtf1, NcSUB1, and Ffcsm1, while TgGATA7 process the unique amino acid sequences of the Zn finger loop (Fig. S2F). As a result, TgGATA6 may be classified into subgroups VI, and TgGATA7 may have a new function in T. guangdongense.

Genes that respond to light can be grouped into two classes: early and late light-induced genes. As a first response, light-activated White-Collar Complex (WCC) binds to the promoters of early light-responsive genes, such as frq, vivid, and sub- 1, to transiently induce or repress their expression (Yu & Fischer, 2018). In the present study, the expression level of TgGATA1 slightly increased when exposed to light for 15 min, but decreased when exposed to light from 30 min to 4 h, suggesting that the expression level of WC1 may peak within the first 30 min. This phenomenon after light treatment for 30min was different from that in Tuber borchii, C. militaris and O. sinensis, in which WC1 was significantly up-regulated within 30 min of light treatment (Ambra et al., 2004; Yang, Xiong & Dong, 2013; Yang & Dong, 2014). Besides, the expression levels of TgGATA5 and TgGATA6 were increased significantly within 30 min of light treatment, suggesting that they may function as early light-induced genes to regulate the downstream target genes. After the first wave of the gene induction, the WCC and other transcriptional factors activate the expression of the late light-responsive genes to regulate the expression of downstream genes (Yu & Fischer, 2018). In this study, TgGATA7 displayed a significant increasing expression pattern after exposure to light for 4 h, suggesting that this gene may be a late light-induced gene or light-induced downstream target gene.

Previous studies have shown that fungal GATA-TFs are involved in sexual and asexual development, and various metabolic processes (Rodriguez-Romero et al., 2010; Yang et al., 2016). In this study, all the seven GATA-TFs in T. guangdongense showed differential expression patterns during fruiting body development. Our results suggested that TgGATA1 (WC1) was notably involved in the mycelia color shift, while TgGATA2 (WC2) was involved in the fruiting body development. TgGATA3 was up-regulation at the mature fruiting body stage (FM), suggesting that it may be related to metabolic processes, such as pigment metabolism. TgGATA4 and TgGATA5 were down-regulated at stages of fruiting body development, suggesting that they may negatively regulate the sexual development. In contrast, TgGATA6 and TgGATA7 were up-regulated at stages of fruiting body development, suggesting that they may positively regulate the sexual development. However, these results are only the preliminary findings with respect to the relationship between TgGATAs and sexual development. The specific functions of these genes require further investigation through genetic approaches.

Conclusions

In this study, seven GATA-TFs were identified in T. guangdongense. TgGATA1 and TgGATA2 can be considered photoreceptors based on their phylogeny, conserved domains, and expression patterns. It was considered that TgGATA3 and TgGATA5 may be involved in nitrogen metabolism and siderophore biosynthesis, respectively, based on the results of motif and homologous analyses. Three genes (TgGATA5-7) were significantly induced by light, while all TgGATAs were involved in fruiting body development to some extent. However, how these genes regulate the fruiting body development should be further analyzed by gene knockout or other genetic approaches. The present results provide comprehensive information on fungal GATA-TFs and lay the foundation for further functional studies on TgGATAs.

Supplemental Information

Table S1 Primers used in this study for qPCR analyses

Click here for additional data file.

Table S2 Amino acid sequences and chromosome positions of TgGATAs

Click here for additional data file.

Table S3 Detailed information of homologous proteins of TgGATAs

Click here for additional data file.

Table S4 Intron-exon structures in TgGATAs

Click here for additional data file.

Table S5 The numbers of amino acid and introns position in TgGATAs and their homologs in Ascomycota

Click here for additional data file.

Table S6 List of the identified motif and their characteristics identified by MEME

Click here for additional data file.

Table S7 List of the predicted cis-elements and their functions

Click here for additional data file.

Table S8 Stability analysis of eight candidate reference genes for qPCR analysis in Tolypocladium guangdongensis under different light conditions calculated by the GeNorm and NormFinder analysis, respectively

Click here for additional data file.

Table S9 Pairwise Pearson correlation coefficients of TgGATAs analyzed by WGCNA according to the previous RNA-seq data

Click here for additional data file.

Figure S1 Motif sequences in TgGATAs and other known functional GATA-TFs

Click here for additional data file.

Figure S2 Amino acid alignments of the Zn finger loops in TgGATA1-TgGATA5 and their homologous proteins

Click here for additional data file.

Additional Information and Declarations

Competing Interests

Author Contributions

Data Availability

The authors declare there are no competing interests.

Chenghua Zhang conceived and designed the experiments, performed the experiments, analyzed the data, prepared figures and/or tables, and approved the final draft.

Gangzheng Wang analyzed the data, authored or reviewed drafts of the paper, and approved the final draft.

Wangqiu Deng performed the experiments, analyzed the data, authored or reviewed drafts of the paper, and approved the final draft.

Taihui Li conceived and designed the experiments, authored or reviewed drafts of the paper, and approved the final draft.

The following information was supplied regarding data availability:

The raw measurements are available in Tables S1-S9.

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
