# Peer review of "Distribution, evolution and expression of GATA-TFs provide new insights into their functions in light response and fruiting body development of Tolypocladium guangdongense"

_PeerJ, doi:10.7717/peerj.9784_

## Round 0.1 · original submission · Major Revisions

Although the reviewers pointed out that you have performed the expected bioinformatic analyses to characterize and classify GATA-TFs of Tolypocladium guangdongense and carried out a comprehensive overview of fungal GATAs, they also reported that the experimental evidence, based on expression data on the effect of light or developmental stage on mRNA levels, is still insufficient for solid conclusions. Therefore, I recommend that you reply point-to-point to the Reviewers’ questions and comments, change the text accordingly, and submit a Revised Version in order to be evaluated again.

Reviewer 1 ·

Basic reporting

Summary
The authors present a study on the GATA-type transcription factors in the fungus Tolypocladium guangdongense partly based on transcriptomics data from a previous study (Zhang et al., 2019). They perform basic bioinformatic analyses of the GATA factors (gene structure, conserved motifs, cis-elements and phylogenetic analysis) in order to classify them in accordance to the already well-described GATAs from species such as Neurospora crassa and Aspergillus nidulans. The authors also investigate the potential light regulation of TgGATAs and their involvement in fruiting body development via qPCR-based expression analyses. While the current state of knowledge about the fungal GATAs is nicely summarized and a good overview of the respective literature is given, the small amount of experimental evidence hardly allows for speculations about the functions of the newly identified TgGATAs. Taken together, the study suffers from several major shortcomings discussed in the following sections.

Introduction
The introduction gives a good overview over the functions of GATA transcription factors in fungi but fails to convey the importance of the analysis in the organism of choice, T. guangdongense by simply describing it as an “edible-medicinal fungus” (line122). I suggest that the authors give some information on the classification and phylogenetic position of T. guangdongense in order to elaborate its importance as model organism for their study.

Figure 1
The alignment of zinc finger domains needs some improvement. I recommend that the authors use a monospaced font like Courier in order to avoid certain letters being cropped. I furthermore suggest mentioning the threshold for consensus highlighting in the figure’s description and adding the position of the zinc finger domains. The authors should also consider including the GATA domain classification, e.g. type IV a and type IV b.

Figure 2
The authors only show one GATA ortholog in S. cerevisiae whereas the literature (e.g. Cooper, 2002; Park et al., 2006) mentions at least ten GATA factors in yeast. I assume they want to express that neither of these are functional/structural orthologs to GATA1-7, but this is still highly misleading to readers unfamiliar with the details of fungal GATA factor classification. In my opinion, the authors should find a way to include all GATA factors for the respective found in literature and databases such as FTFD. Adding a sub-group for non-orthologues GATA factors such as the ones found in yeast would help understanding the full complexity of GATA factor classification and subdivision in fungi.
Cooper, T. G. (2002). Transmitting the signal of excess nitrogen in Saccharomyces cerevisiae from the Tor proteins to the GATA factors: connecting the dots. FEMS microbiology reviews, 26(3), 223-238.
Park, J. S., Kim, H. J., Kim, S. O., Kong, S. H., Park, J. J., Kim, S. R., ... & Lee, Y. H. (2006). A comparative genome-wide analysis of GATA transcription factors in fungi. Genomics & Informatics, 4(4), 147-160.


Figure 3
The phylogeny unambiguously shows the GATA factor subdivision and is perfectly valid in that sense. The authors should restrain from drawing “evolutionary” (line 250) conclusions from a NJ phylogeny since it is merely based on a distance matrix rather than an evolutionary model. I suggest the use of maximum likelihood or Bayesian methods, which use models of sequence evolution, if they want to speculate on GATA factor evolution in fungi. If they simply want to subdivide the GATA factors based on sequence similarity NJ is acceptable, but this should be stated in the method section.

Figure 6
The authors should give information on the phylogeny construction and name the already well characterized motifs, e.g. zinc fingers and PAS domain, in the figure’s legend. I recommend also performing domain identification on the so far uncharacterized motifs in table S5 via e.g. Prosite and annotate them as far as possible.

Figures 8 + 10
I suggest separating the plots not by gene but by duration of light treatment (Fig. 8)/developmental stage (Fig. 10) or showing the data in grouped bar chart in order to facilitate comparing the expression of the different genes at the same time point.
In lines 447 – 448 the authors assume that WC1 expression “may peak within the first 30 min” while WC1/TgGATA1 still seems to be downregulated at 30min and expression is increasing from 2h to 4h of light treatment in Fig. 8.

Materials & Methods
I suggest using [µmol photons m-2 s-1] rather than [lux] for light intensity.
The authors should also provide the light conditions for fruiting body development.

qPCR-based expression analyses (Table S7):
The authors use amplicons drastically different in size, ranging from 158 bp for CCG_01872 up to 474 bp for CCG_00651, and still assume constant amplification efficiency for all samples in using Livak’s 2-ΔΔCT. Bustin and Huggett have shown that longer amplicons are detected earlier in SYBR Green qPCR assays. I therefore suggest that the authors use a variation of the Model based analysis of real-time PCR data (Alvarez et al., 2007) described in Grimes et al., 2014.
Bustin, S., & Huggett, J. (2017). qPCR primer design revisited. Biomolecular detection and quantification, 14, 19-28.
Alvarez, M. J., Vila-Ortiz, G. J., Salibe, M. C., Podhajcer, O. L., & Pitossi, F. J. (2007). Model based analysis of real-time PCR data from DNA binding dye protocols. BMC bioinformatics, 8(1), 85.
Grimes, B. T., Sisay, A. K., Carroll, H. D., & Cahoon, A. B. (2014). Deep sequencing of the tobacco mitochondrial transcriptome reveals expressed ORFs and numerous editing sites outside coding regions. BMC genomics, 15(1), 31.


Comments on language, grammar and misc.
Line 107: “Light is an important environmental signal for their sexual and asexual growth, circadian …“
Lines 436 – 438: The authors use the word “process” for what I assume should be “possess”.
Line 480: The authors identified seven GATA orthologs, not seven groups of GATAs.
Lines 468 – 469: Missing citation for involvement of fungal GATAs in sexual and asexual development.

Possible Experiments to further investigate the role of TgGATAs
• Expression analysis of TgGATAs in response to blue/red/white light
• Identification and expression analysis of orthologs of known downstream targets of NcWC1/2 in response to different light signals
• Generation of TgGATA k.o. mutants in order to verify the assumptions about the involvement in fruiting body development

Experimental design

see above

Validity of the findings

see above

Additional comments

see above

Reviewer 2 ·

Basic reporting

The manuscript “Distribution, evolution and expression of GATA-TFs provide new insights into their functions in light response and fruiting body development of Tolypocladium guangdongense”by Zhang and co-workers describes the systematic identification and characterization of GATA-TFs in T. guangdongense. The authors use a pool of pre-defined software tools that have become a standard for similar studies over the past years. Finally, they conclude by identifying putative GATA TFs that play a crucial role in light response and fruiting body development in T. guangdongense.
The manuscript is mostly well written with a little tweaking of the English language required. Nevertheless, I am not a native English speaker and it would be a good idea to get advice from a native English speaker.
In general, the introduction is well structured gradually gliding the readers to essential background information with appropriate literature citations and the need for undertaking the present study. The figures are well illustrated with descriptive legends and are of publication quality. Appropriate raw data has been provided.

Experimental design

The authors describe a regular meta-analysis for identifying and characterizing GATA TFs in T. guangdongense which falls well in the scope of the PeerJ journal. The chosen topic of the study seems justified as there are limited studies describing the characterization of GATA-TFs in T. guangdongense. The methodology adopted is widely used for identifying and characterizing TFs and is described adequately for other researchers to replicate the analysis. The results are described in enough detail and interpreted well. The discussion section is well constructed with a clear balance between the interpretation of the results obtained and associated speculations.
Few minor points:
1. How many numbers of replicates were used for qPCR expression analysis: biological and technical replicates? Also, were the results statistically significant?
2. The authors state the availability of transcriptome data for different light conditions. It would be nice to compare the GATA expression profiles obtained from RNASeq and qRT-PCR.
3. Please cite the qRT-PCR primer table in the method section as this section appears first.

Validity of the findings

The present study is preliminary in nature and is identified as such by the author which is appreciable. The exact functional mechanisms of the GATA TFs identified are far from complete which requires concrete functional studies involving knock out/overexpression. I hope the authors will venture into such studies in the near future to come up with meaningful conclusion of their present findings.

Additional comments

I commend the exhaustive analysis performed by the authors and recommend the publication of the manuscript after a minor revision.

Reviewer 3 ·

Basic reporting

The manuscript "Distribution, evolution, and expression of GATA-TFs provide new insights ...", by Zhang et al, describes a detailed sequence analysis of the seven GATA TFs found in the genome of the edible-medicinal fungus Tolypocladium guangdongense. Putative functions of the genes are attributed according to the similarities to well-investigated orthologs in other fungi, in agreement with their sequence features. The work is very weak: a major body of results in the manuscript resides in sequence analysis and bioinformatic predictions, making many of the conclusions too speculative. The results are combined with expression data on the effect of light or developmental stage on mRNA levels of the seven genes, measured by RT-PCR. In general, the results are insufficient for solid conclusions.

Experimental design

I have several major concerns:

1) About the mycelia age used to illuminate the samples: 20 days seems a too long incubation time for a fungus. Why do the authors use old mycelia to be illuminated? Experiments with other fungi usually used in research, including Neurospora, are carried out with much younger mycelia. I assume that the mycelia used in the illumination experiments are in a very late stationary phase. Why was this time chosen? Have experiments to investigate the effect of mycelial age on the response An experiment is missing to look for a more appropriate incubation time.

2) It is not clear the reason to use the vacuolar protein sorting (VPS) as a single reference control gene in the RT-PCR studies. Is it really a constitutively expressed gene under the conditions investigated? It is strange that the genes for such different GATA factors, involved in very different metabolic aspects, have similar regulatory features. Changes in mRNA levels for the investigated GATA genes are not very strong (induction level after illumination, for example, is much higher in N. crassa for the wc-1 gene). E.g, the transient reduction of mRNA levels after illumination that affects similarly to different GATA genes might be explained by a minor increase in the VPS mRNA levels in the same conditions. It is necessary to use at least a second reference gene. Usually, house-keeping genes, like those for actin or beta-tubulin, are normally used in RT-PCR quantifications. Which is the basis to use VPS?

3) Experimental conditions are so poorly explained. Please, note that descriptions should allow other researchers to repeat the experiments under the same conditions. A separate section for strains (the strain is not indicated!) and culture conditions should be indicated. How were inoculations done? The samples for the developmental conditions, how were they taken? Are they supposed to be in liquid culture? Do the authors take samples from specific structures or the total mycelial samples? With the provided information it is not possible to reproduce the experiments.

4) I have doubts about the co-expression analysis. I do not doubt on the usefulness of mathematical or bioinformatics resources, such as the WGCNA, but the results should be consistent with what the reader may visually grasp. In the graphs displayed in Fig. 8, I see more correlation in the expression pattern between GATA2 and GATA3 than between GATA1 and GATA2. However, in Fig. 9 GATA2 and GATA3 are linked by a green line. Is is also strange the red line for GATA5 and GATA6 in Fig. 9, which however show quite different patterns in Fig. 8. I am not very confident of an approach that goes against what your eyes perceive.

5) It is stated in the figure legends of RT-PCR experiments that the error bars indicate standard deviation of three experiments. Do the authors mean technical repetitions or biological repetitions? An extremely low variation is observed in three experiments for some of the genes (see, e.g., GATA 1 and GATA 6 in Fig. 8, or GATA1 and GATA 7 in Fig. 10). In my experience, RT-PCR data are always subject to biological noise, which is added to technical noise, which makes these extremely low variations quite unusual.

Validity of the findings

Bioinformatic findings may be reliable, but without experimental support, they are just preliminary clues awaiting for experimental demonstration. As indicated in section 1, irrespective of the experimental concerns, available data are very insufficient for solid conclusions.

Additional comments

In general, I find this work too preliminary and speculative. I have no doubts about the reliability of the bioinformatic analyses, which are very rigorous and exhaustive, but at the same time, they are to a large extent very conventional and straightforward to achieve, in many cases consisting just on the output of online tools. On the other hand, experimental work is not sufficiently explained and reliability is more doubtful.
The weakness of the results makes the conclusions too speculative. This is especially visible in the paragraph of lines 441-457 in the discussion. The data are too poor to establish possible regulatory connections between different GATA genes. In addition, there are sentences that make no sense. E.g.: " ...transcripts of TgGATA1 increased as light treatment increased from 30 min to 4 h, speculating that the expression level of WC1 may peak within the first 30 min..." Within the first 30 min does not fit with an increase from 30 min to 4 h. I do not understand also the logic of the following sentence "Besides, the expression levels of TgGATA4, TgGATA6, and TgGATA7 were also down-regulated, speculating that they may function as early light-induced genes to regulate the downstream target genes". Downregulation is transient and rather small, but, why does it mean that "they may function as early light-induced genes"?

Some minor comments that may help the authors for future resubmissions:

Line 32. "Seven subgroups of TgGATAs were identified". The same idea is found in the conclusion (line 480). The authors mean seven TgGATA genes, not subgroups.

Lines 71-72. WC-2 is not a photosensitive protein. In the WC photoreceptor complex, the only photosensitive protein is WC-1.

Line 94, People working with S. cerevisiae usually ignore that there are plenty of different yeasts in nature. Better to say S. cerevisiae.

Line 226: chromosomes have no orientation, it is just arbitrary and, in any case, it is irrelevant information unless referred to closely linked genes.

Line 397: "proteins of GATA1 and GATA2 are rarely found in S. cerevisiae due to lowest identities" The authors mean that they are not found. Also the next sentence "It is considered that photoreceptors, like WC-1 and WC-2, may not exist in S. cerevisiae". It is well known that S. cerevisiae lacks light responses and actually lacks WC photoreceptors.

Line 436: "...that TgGATA6 process similar..." should be "...that TgGATA6 possess similar... "

---

## Round 0.2 · Minor Revisions

Two of the reviewers still asked for minor revisions that must be attended and both of them suggest that the English writing must be polished. Thus, the manuscript still needs a number of Minor Revisions.

Reviewer 1 ·

Basic reporting

I think the English can still be improved here and there but is much better than in the previous version. (Comment to reviewer #3, line 397: "....photoreceptors are not exist in this fungus (Part et al., 2016).

Experimental design

The revised version is much improved and all my comments have been addressed.

Validity of the findings

The findings are still highly speculative and there is not much depth in this paper. However, all the criticisms to the existing paper have been addressed.

Additional comments

Please revise the use of the English language once more.

Reviewer 2 ·

Basic reporting

The authors have adequately addressed all the queries raised by the reviewers.

Experimental design

no comment

Validity of the findings

no comment

Additional comments

Although no additional studies have been conducted to improve the functional role of GATA TFs, the manuscript has been revised adequately with an appropriate response to the queries raised and hence could be accepted for publication in PeerJ journal.

Reviewer 3 ·

Basic reporting

The authors have made a considerable effort to improve the manuscript and have generally responded satisfactorily to the reviewers.
In overall, the manuscripts dealt in big detail with the subject of the GATA Proteins in Tolypocladium, especially at the in silico level. Data on expression, although reliable, must be considered only preliminary from the point of view of functional assignations, but they are now technically more sound than in the former submission and are worthy of publication. Data are in general reasonably described, discussed, and referenced, and figures and tables have been carefully done.
I am not a native english speaker, and therefore I am no competent to judge grammar aspects.
About manuscript content, I only suggest some minor changes. I consider the manuscript in its current state to be an interesting contribution to the understanding of GATA family of proteins and I consider that only minor improvements are necessary before publication.

Experimental design

In my opinion, the issue of mycelial age or illumination conditions has not received enough attention in this research. However, there are too many data obtained with the present growth conditions, and so the present work should stay in its current form. However, I recommend the authors to consider for future works to carry out time course experiments to check the effect of age, or try alternative growing conditions. For example, if colonies grow very slowly, many colonies can be inoculated in the same petri dish, or cultures can be grown by spreading spores on agar surface to obtain a mycelial mat.

It is also a weak aspect to use the entire mycelium at the different stages of sexual maturation when this only occurs at well delimited structures in the mycelium. If development occurs at localized points, the data are logically mixed with those of mycelium that does not participate in these phenomena and might be subject of other regulatory scenarios. It is difficult to interpret the data without having information on how abundant are such structures or how much do they represent over the total mycelial mass.
These are possible improvements to take into account in future work, but the data in the present study should be already sufficient for publication.

Validity of the findings

I still have my doubts about the functional extrapolations from expression data, especially when differences are not so drastic. It should be borne in mind that regulatory proteins control the expression of other genes, and often respond to signals with post-transcriptional changes, and do not necessarily have to be importantly regulated at their mRNA level. As an example, the Neurospora Wc-1 gene is regulated by light (actually, self-regulated), but its ortholog in Fusarium is not, despite the fact that both have similar functions.
Thus, interpretations must be especially cautious. Eventually, the verb "indicate" should be rather replaced by "suggest" in some places. See. e.g., in lines 630-632: "TgGATA4 and TgGATA5 were down-regulated at stages of fruiting body development, indicating that they may negatively regulate the sexual development", better to say "TgGATA4 and TgGATA5 were down-regulated at stages of fruiting body development, SUGGESTING that they may negatively regulate the sexual development"

Additional comments

As a major comment to the authors, thanks for the efforts to improve the expression data, which are now more reliable.

Please, consider some improvements in the data:

Data in Table 2: there are some species with additional GATAs. Could they be in some case paralogs of the "official" GATAs?
An observation about the display of data in the table: the symbols chosen are too similar for a clear distinction. It would be visually more clear if, e.g., colored circles are used for the positives. Also, please note that pale colors (typically yellow) are not very visible, especially when printed in black and white.

Bars for dark conditions in Fig. 7 are not real and can be removed; it must be indicated that values are referred to those in the dark, that were taken as 1. Displayed in that way, it leads to the misleading impression that all genes are expressed equally in darkness. Actually, it would be a good idea to provide information on real absolute values in the dark, e-g., from RNA-seq data.
The same can be said for the mycelium values in Fig. 8.

Some improvements in the text:

Lines 156-158: The sentence "To identify the TgGATAs, the online analysis platform, InterProScan (Zdobnov and Apweiler, 2001) was used to select the proteins with conserved GATA-domains by scanning the T. guangdongense protein database" should rather be "To identify the TgGATAs, the online analysis platform, InterProScan (Zdobnov and Apweiler, 2001) was used to screen the T. guangdongense protein database for proteins with conserved GATA-domains"

Lines 392-394: the newly added sentence "Based on the results of reference genes selection, α-tub1 and β-tub1 were selected as reference genes for analysis the relative expression levels of TgGATAs" should rather be "Based on the results on reference genes, α-tub1 and β-tub1 were selected as reference genes for analysis of relative expression levels of TgGATAs"

Line 536: "regulatory connections may be exist between..." should be "regulatory connections may exist between..."

Line 538: "may be a downstream target gene" rather than the ""may be the downstream target gene"·

---

## Round 0.3 · accepted · Accept

All the questions raised by the reviewers in this second round were sufficiently answered and the article was modified accordingly. Furthermore, the authors performed a professional English editing in the text, as suggested by reviewers; Thus, I am very pleased to inform that your article has been accepted!